

# Physical and chemical properties of urban aerosols in São Paulo, Brazil: Links between composition and size distribution of submicron particles.

Djacinto Monteiro dos Santos[1], Luciana Varanda Rizzo[2], Samara Carbone[3], Patrick Schlag[1a], and Paulo Artaxo[1]

[1]Institute of Physics, University of São Paulo, São Paulo, Brazil
[2]Instituto de Ciências Ambientais, Químicas e Farmacêuticas, Universidade Federal de São Paulo, Diadema, São Paulo, Brazil
[3]Federal University of Uberlândia, Uberlândia, Minas Gerais, Brazil
[a]now at: Shimadzu Deutschland GmbH, 47269 Duisburg, Germany

**Correspondence:** Djacinto Monteiro dos Santos (djunior@if.usp.br)

**Abstract.** In this work, the relationships between size and composition of submicron particles ($PM_1$) were analyzed at an urban site in the Metropolitan Area of São Paulo (MASP), a megacity with about 21 million inhabitants. The measurements were carried out from $20^{th}$ December 2016 to $15^{th}$ March 2017. The chemical composition was measured with an Aerodyne Aerosol Chemical Speciation Monitor and size distribution with a TSI Scanning Mobility Particle Sizer 3082. $PM_1$ mass

concentrations in the MASP had an average mass concentration of 11.4 $\mu$g m$^{-3}$. Organic aerosol (OA) dominated the $PM_1$ composition (56%), followed by sulfate (15%) and equivalent black carbon (eBC, 13%). Four OA classes were identified using Positive Matrix Factorization: oxygenated organic aerosol (OOA, 40% of OA), biomass burning organic aerosol (BBOA, 13%), and two hydrocarbon-like OA components (a typical HOA related to vehicular emissions (16%), and a second HOA (21%) representing a mix of anthropogenic sources). Particle number concentrations averaged 12100 $\pm$ 6900 cm$^{-3}$, dominated by

the Aitken mode. Accumulation mode increased under relatively high $PM_1$ concentrations, suggesting an enhancement of secondary organic aerosol (SOA) production. Conversely, the contribution of nucleation mode particles was less dependent on $PM_1$ levels, coherent with vehicular emissions. The relationship between aerosol size modes and $PM_1$ composition was assessed by multilinear regression models. Mass loading in the nucleation mode was associated mostly with eBC, HOA and OOA, suggesting the contribution of primary and secondary particles from traffic sources. Secondary inorganic aerosols were

partitioned between Aitken and accumulation modes, related to condensation particle growth processes. Submicron mass loading in accumulation mode was mostly associated with highly oxidized OOA and also traffic-related emissions. To the author's knowledge, this is the first work that use the MLR methodology to estimate the chemical composition of the different aerosol size modes. The results emphasize the relevance of vehicular emissions to the air quality at MASP and highlight the key role of secondary processes on the $PM_1$ ambient concentrations in the region since 56% of $PM_1$ mass loading was attributed to SOA

and secondary inorganic aerosol.





## 1 Introduction

Air pollution has become a major problem in large urban conglomerates, particularly in megacities with a population exceeding 10 million inhabitants (Fuzzi et al., 2015). At these megacities, industrial and vehicular emissions are generally the main air pollution sources (Beirle et al., 2011; Zhu et al., 2012). The Metropolitan Area of São Paulo (MASP) is the largest South
American megacity, with a population of about 21 million inhabitants (IBGE, 2020), an area of 7,947 km$^2$, comprising 39 municipalities and a fleet of 8 million vehicles. An important characteristic of the Brazilian vehicle fleet is that gasohol with 24% ethanol and pure ethanol are the main fuels, making Brazil, and especially the MASP, a unique case of large-scale biofuel usage worldwide. Successful public policies for controlling and monitoring industrial and vehicular pollutant sources have been implemented over the last decades in the MASP leading to positive impacts in reducing the emission of primary pollutants to
the atmosphere (Andrade et al., 2017). However, many uncertainties are still found on the effect of emission reduction policies on the concentration of secondary pollutants, for example, ozone and secondary aerosols.

Aerosol particles have recognized effects on human health (Cohen et al., 2017) and critical role in the Earth's radiation budget (Boucher et al., 2013). However, the knowledge of the dynamics of physicochemical properties of submicron particles remains limited. The chemical composition and the size distribution of ambient aerosol are key parameters concerning its
optical properties (Romano et al., 2019), ability to act as cloud condensation nuclei (CCN) (Che et al., 2016) and deposition efficiency into the human respiratory tract (Ching and Kajino, 2018). Furthermore, aerosol properties range widely in the atmosphere as a result of interaction between local and regional sources (Costabile et al., 2009) and atmospheric processing (Jimenez et al., 2009). Therefore, a current scientific effort is the characterization of the mechanisms of aerosol production and aging under ambient conditions, in particular the formation of secondary organic aerosols (SOA).

Source apportionment studies carried out at MASP (Castanho and Artaxo, 2001; Andrade et al., 2012; Pereira et al., 2017; de Miranda et al., 2018) have shown that vehicle traffic, resuspended soil dust and secondary processes are the main sources of PM$_{2.5}$ in the region. Although such studies have provided innovative knowledge about the main air pollution sources in MASP, most of them derived from filter-based offline chemical analyses. Low time resolution offline measurements are unable to describe the dynamic of the sources and processes of urban aerosols. To our knowledge, Brito et al. (2018) were the only
ones to report high-resolution measurements of chemical composition of submicron particles recently in the MASP, so that conducting further studies in the region is fundamental for a better understanding of such dynamic processes.

Herein, near real-time PM$_1$ chemical composition was evaluated, including black carbon, inorganic aerosols and chemical classes of organic aerosol, using Positive Matrix Factorization. Furthermore, the relationships between aerosol chemical composition and particle number size distribution (PNSD) were investigated by using a multilinear regression (MLR) approach.
The results presented here provide innovative insights on the association between sources and processes governing physical-chemical properties of atmospheric aerosol in a megacity largely impacted by traffic emissions and extensive biofuel usage.





## 2 Methodology

### 2.1 Sampling site and measurements

An aerosol and trace gas monitoring station was operated at one of the municipalities of the Metropolitan Area of São Paulo
(MASP), at the Federal University of São Paulo (UNIFESP, 23º43'8"S 46º37'40"W, 769 m above mean sea level), in Diadema
city (Figure 1). The municipality of Diadema is located 20 km southeast from São Paulo downtown in a region characterized
by high population density and strong vehicular impact, besides emissions from industrial activities and proximity to preserved
Atlantic forest areas. The combination of different anthropogenic and biogenic emissions results in complex physicochemical
processes that promote the formation of SOA and ozone.

The measurements were carried out from $20^{th}$ December 2016 to $15^{th}$ March 2017, comprising 105 days of data acqui-
sition. Trace gas analyzers Thermo 49i and Thermo 43i were used to monitor the concentration of trace gases $O_3$ and $SO_2$,
respectively. $NO_2$ concentration was monitored using a Cavity Attenuated Phase Shift analyzer (CAPS, Aerodyne Research
Inc.). Submicron particle number size distributions (PNSD) in the range 10-450 nm were measured every 2 minutes, using TSI-
SMPS Model 3082, associated with a TSI-CPC Model 3772. Particle absorption coefficients were measured at 637 nm using a
Thermo Scientific Model 5012 MAAP (Multi-Angle Absorption Photometer), compensated by a +5% factor and converted to
equivalent black carbon (eBC) mass concentration assuming a mass absorption efficiency value of 6.6 $m^2$ $g^{-1}$ (Müller et al.,
2011).

The chemical composition of non-refractory submicron aerosol was measured using a Quadrupole Aerosol Chemical Spe-
ciation Monitor (Q-ACSM, Aerodyne Research Inc.), described in detail by Ng et al. (2011a). Briefly, aerosol particles are
sampled using a $PM_{2.5}$ cyclone inlet and focused through an aerodynamic lens forming a narrow particle beam, which is
transmitted into the detection chamber under high vacuum onto a 600°C vaporizer. Non-refractory material is flash-vaporized
and ionized by electron impact at 70 eV. According to their mass-to-charge ratio (m/z), the fragments are analyzed in the
quadrupole spectrometer. In this study, an averaging interval of 30 min was adopted. The collection efficiency (CE, defined as
the ratio of the mass of particles detected by the instrument to the mass of particles introduced in the inlet) was calculated using
the parametrization of Middlebrook et al. (2011) and resulted in an average of 0.45 ± 0.01. The transmission of the ACSM lens
makes the instrument measure particles from 70 nm to about 900 nm, normally referred as $PM_1$. Detection limits of ACSM
were calculated as three times the average noise level, using data of ambient air sampled through a HEPA filter. For 30 min of
averaging time, the $3\sigma$ detection limits for chloride, ammonium, nitrate, organics, and sulfate are 0.03 $\mu g$ $m^{-3}$, 0.36 $\mu g$ $m^{-3}$,
0.03 $\mu g$ $m^{-3}$, 0.57 $\mu g$ $m^{-3}$, and 0.09 $\mu g$ $m^{-3}$, respectively.

Ambient aerosols were sampled under dry conditions (RH<40%) using a nafion dryer, and concentrations were compen-
sated for standard temperature and pressure conditions (1013.25 mbar; 273.15 K). Aerosol and trace gas measurements were
averaged in periods of 30 min. Meteorological parameters (wind speed, wind direction, temperature, RH, solar radiation and
precipitation) were provided by the Institute of Astronomy, Geophysics and Atmospheric Sciences of the University of São
Paulo (IAG) meteorological station (23°39'02.61"S 46°37'18.55"W), 10 km North from the sampling site.





## 2.2  Identification of OA components with Positive Matrix Factorization (PMF)

Positive Matrix Factorization (PMF) is a statistical model that uses weighted least-square fitting for factor analysis (Paatero and Tapper, 1994; Paatero, 1997). It uses a bilinear factor analytic model defined in matrix notation as:

$$X = GF + E \tag{1}$$

where X denotes the matrix of the measured values, G and F are matrices computed by the model that represent the scores and loading, respectively, and E is the residual matrix, made up of the elements $e_{ij}$. For the ACSM and AMS (Aerosol Mass Spectrometer) data, the measured organic mass spectra are apportioned in terms of source/process-related components (Zhang et al., 2011). In this case, the columns j in X are the m/z's and each row i represents a single mass spectrum, G represents the time series and F the profile mass spectrum for the p factors computed by the algorithm. The model adjusts G and F using a least-squares algorithm that iteratively minimizes the quantity Q, defined as the sum of the squared residuals weighted by their respective uncertainties:

$$Q = \sum_{i=1}^{m} \sum_{j=1}^{n} \left( \frac{e_{ij}}{\sigma_{ij}} \right)^2 \tag{2}$$

where $\sigma_{ij}$ is the uncertainty for each element in the matrix X. An IGOR™-based source finder (Canonaco et al., 2013, Sofi) with a multilinear engine algorithm (Paatero, 1999, ME-2) was used to prepare the data, error estimates, execute the analysis and evaluate the results.

## 2.3  Multilinear regression (MLR) model

An analysis of the relationship between chemical composition and size distribution was performed using a multilinear regression (MLR) model. Previous studies have applied the MLR model to estimate aerosol mass scattering and extinction efficiencies, and source apportionment of optical properties (Ealo et al., 2018). A linear regression model describes the relationship between a dependent variable, y, and one or more independent variables, x. The dependent variable is also called the response variable and independent variables are also called explanatory or predictor variables. The MLR model is:

$$y_i = \beta_0 + \beta_1 x_{i1} + \beta_2 x_{i2} + ... + \beta_p x_{ip} + \epsilon_i, i = 1, ..., n \tag{3}$$

where, $y_i$ is the $i^{th}$ observation of the response variable, $\beta_j$ is the $j^{th}$ coefficient, $\beta_0$ is the constant term in the model, $x_{ij}$ is the $i^{th}$ observation on the $j^{th}$ predictor variable, j = 1, ..., p, and $\epsilon_i$ is the $i^{th}$ error term.

For the MLR model, the time series of $PM_1$ chemical constituents (i.e., eBC, inorganic species and OA PMF-derived chemical classes) were used as dependent variables, and the volume of particle size modes (i.e., nucleation, Aitken and accumulation) were taken as predictors. Volume size distribution was used in all MLR calculations since it represents better the accumulation





and Aitken modes than number size distribution. For MLR analysis, intercept (constant term in the model) was set to 0, as it holds no physical significance for the mass attribution. Since the presence of multicollinearity (i.e. correlation between predictors) can adversely affect results in an MLR, the variance inflation factor (Hair et al., 2006, VIF) was determined. The VIF is a means of detecting multicollinearities between the independent variables of a model. If the VIF > 10, the multicollinearity is high and the variable should be excluded (Hair et al., 2006). A fit linear regression model was performed using fitlm function for Matlab 2015a by using standard least-squares fit.

## 3 Results and Discussion

### 3.1 Meteorology and trace gases

The campaign was carried out during the Southern Hemisphere spring/summer, with an average ($\pm$ standard deviation) air temperature of $23.0 \pm 3.5°C$, varying from 14 to 33°C. RH was $79.6 \pm 15.7\%$, on average (Table 1). The monthly accumulated precipitation ranged between 140 and 370 mm. The wind rose (Fig. 2) indicates dominant winds from the northeast, northwest and southeast along the measurement period.

Average mixing ratios of trace gases are shown in Table 1. The mid-day $O_3$ (averaged between 10h-18h LT) peak was 43.5 ppb, values comparable to the measurements at the Ibirapuera Park, one of the areas of MASP that often exceeds ozone air quality standards (Andrade et al., 2017). Diadema site is surrounded by green areas, an important contributor source of biogenic volatile organic compounds (BVOC), adding to VOC emissions from anthropogenic sources (Brito et al., 2015), in promoting the formation of ground-level ozone and SOA. In the MASP, the ozone levels are more sensitive to changes in VOC emissions rather than NOx (VOC-limited), being the only pollutant that shows an increasing trend for concentrations in the MASP over the years due to the large use of fossil fuel and biofuels (Andrade et al., 2017).

The average $NO_2$ concentration in this study was 13 ppb, significantly lower compared to São Paulo downtown (Andrade et al., 2017). According to emission inventories (CETESB, 2019), vehicles account for 64% of $NO_X$ emissions in the MASP, particularly diesel engines in HDVs (heavy-duty vehicles: mostly buses and trucks), which represent 44% of total $NO_X$ emissions. The average $SO_2$ concentration is 0.61 ppm, approximately half of the observed values in São Paulo downtown during springtime in 2013 (Monteiro dos Santos et al., 2016). Few episodes (15% of the dataset) presented $SO_2$ levels above 1 ppm. In the MASP, $SO_2$ emissions are related to industrial sources and the sulfur content in diesel and gasohol.

### 3.2 Aerosol chemical composition

Near real-time submicron mass concentration ($PM_1$) can be obtained by the non-refractory $PM_1$ ($NR$-$PM_1$) and eBC measurements (Table 2). Average $PM_1 \pm$ standard deviation during the campaign was $11.4 \pm 7.8$ $\mu g$ $m^{-3}$. These results are similar to observations in São Paulo downtown (10.8 $\mu g$ $m^{-3}$, Brito et al. (2018) during the spring), New York (11.7 $\mu g$ $m^{-3}$ Sun et al. (2011) during the summer), Barcelona (18.5 $\mu g$ $m^{-3}$, Mohr et al. (2012) during the winter) and Santiago (18.1 $\mu g$ $m^{-3}$, Carbone et al. (2013) during the spring). On average, organic aerosols dominated the composition, contributing to 55%, fol-





lowed by sulfate (15%) and eBC (14%). Ammonium (9%), nitrate (6%) and chloride (1%) presented smaller contributions to the mass loading. The time series of $PM_1$ chemical species and its relative contribution to the total mass concentration in the

submicron size range is shown in Figure 3.

Components of OA were identified using PMF analysis following the procedure described by Ulbrich et al. (2009). The choice of the number of factors was based on the quality of the fit parameter, the correlation analysis with external tracers, spectral analysis and comparisons with AMS database mass spectra, and their diurnal variability. Moreover, the examination of rotational ambiguity was done by varying the FPEAK parameter. Solutions with more than four factors were examined and

showed only the splitting behavior of existing factors, instead of providing new consistent factors. Details of the PMF analysis procedure are given in the Supplement Material. The identified four OA factors include: a highly oxidized component, linked to secondary processes, named oxygenated organic aerosol (OOA); a component containing typical biomass burning spectral signature, named biomass burning organic aerosol (BBOA); and two hydrocarbon-like OA components ($HOA_I$ and $HOA_{II}$), that represent primary emissions from anthropogenic sources, but with distinct mass spectra and diurnal variability patterns.

The OOA component is dominated by m/z 44, which is mainly the fragment $CO_2^+$ (Zhang et al., 2005), typical from thermal decarboxylation of organic acid groups, previously described as aged aerosol and related to the formation of SOA (Jimenez et al., 2009). The OOA factor is the dominant component, comprising, on average, 40% of OA total mass concentration. The mass spectrum of OOA (Figure 4) correlates strongly with the standard AMS LV-OOA mass spectra database (R=0.99, Table 3), indicating that the OOA factor is dominated by low volatile organic compounds, rather than semi-volatile organic

compounds (R=0.72, Table 3). The OOA time series present moderate correlations with oxidant concentration ($O_X$=$NO_2$+$O_3$, Table 4), and secondary inorganic species, such as sulfate, nitrate and ammonium. Moreover, the OOA mass concentration significantly increases in the afternoon, similarly to the ozone diels (Figure 5), indicating that its formation is partially driven by photochemistry. Considering the sum of secondary inorganic aerosols (sulfate, nitrate and ammonium) and SOA (OOA) as a lower limit for the contribution of secondary aerosols to the total of $PM_1$, it is possible to estimate that at least 56% of

submicron particles mass loading results from secondary production.

The BBOA component has a mass spectrum dominated by the m/z's 29, 60 and 73 (Figure 4). The signal at m/z 60 is associated with the $C_2H_4O^+$ ion (Alfarra et al., 2007) and correlates with levoglucosan and similar anhydrosugar species (mannosan, galactosan) that result from the pyrolysis of cellulose. The BBOA mass spectrum presents a strong correlation with standard AMS database BBOA (R=0.90, Table 3). The diurnal variability of BBOA (Figure 5), with an average concentration almost

three times higher during nighttime, seems modulated by atmospheric dynamics, such as boundary layer height evolution. The boundary layer height decreases during nighttime trapping freshly emitted smoke particles. The time series of BBOA correlates moderately with eBC (R=0.47), nitrate (R=0.48) and chloride (R=0.56) (Table 4). The average contribution of BBOA is 13% of total OA and almost 7% of $PM_1$, significantly lower than reported in Pereira et al. (2017), that found considerable biomass burning contributions (approximately 18% of $PM_{2.5}$) associated to long-range transport from regional sugarcane burning in

São Paulo during wintertime, in addition to local emission sources.

Both HOA components present mass spectra characterized by hydrocarbon-like structures typical of alkanes, alkenes and cycloalkanes (m/z's 27, 29, 41, 43, 55, 57, 67, 69, 71, 81, 83, 85) related to anthropogenic primary emissions (Canagaratna





et al., 2004; Zhang et al., 2005). Both mass spectra correlate with the AMS standard HOA mass spectrum (R=0.90 and R=0.89 for $HOA_I$ and $HOA_{II}$, respectively). Although both factors are HOA related, it is not reasonable to interpret them as a split

of the same source. The $HOA_I$ factor presents an elevated signal at m/z 55 that has been related to cooking OA (COA), an important source of (primary organic aerosol) POA in urban environments (Mohr et al., 2012). The $HOA_{II}$ factor presents a higher signal at m/z 57 than at m/z 55, and higher correlation with eBC ($R_{HOA_{II}}$=0.69, $R_{HOA_I}$ =0.45), which is related to vehicular emissions in the MASP, mostly heavy-duty vehicles (de Miranda et al., 2018). The results indicate that $HOA_{II}$ is more consistent with traffic, while the $HOA_I$ seems like a mixture of anthropogenic sources. Together, the HOA factors present

an average contribution of 37% to OA (21% from $HOA_I$ and 16% from $HOA_{II}$). For both HOA factors, the diurnal profiles of mass concentrations (Figure 5) increase during the traffic rush-hour time 6h-8h (local time). However, $HOA_I$ also shows a peak between 12h and 14h (local time), probably associated with local cooking activities.

Although the sampling site is located in an industrialized region, a distinct industrial-related OA factor could not be identified in this study. As a matter of comparison, Bozzetti et al. (2017) identified OA related to industrial emissions in an urban

background site in Marseille under influence of industries and petrochemical companies. Some similarities can be observed between the mass spectrum of $HOA_I$ in this study and the mass spectrum of industrial-related OA (INDOA) that they found, particularly substantial signals at m/z 27, m/z 29, m/z 43 and m/z 55. This fact can indicate the influence of industrial emissions in $HOA_I$, however, further investigation is necessary. Furthermore, a significant part of ammonium sulfate could be produced from the oxidation of anthropogenic $SO_2$ from industrial sources.

## 3.3 Aerosol size modes

Based on the approach of Hussein et al. (2005), one to three lognormal modes were fitted to each measured PNSD in the range 10-429 nm. Considering the average PNSD, nucleation, Aitken and accumulation modes were centered at mobility diameters of 22 nm, 50 nm and 122 nm, respectively (Figure 6). For nucleation and Aitken modes these results are very similar to previous observations in São Paulo region during spring and early summer (Backman et al., 2012), however, the diameter

for accumulation mode is significantly lower than the reported in such study (>200 nm). The sum of the number concentration of nucleation, Aitken and accumulation modes explains the measured total particle number concentration (slope=0.98 and $R^2$=0.99). The Aitken mode dominated the PNSD with average concentration $\pm$ standard deviation of $6900 \pm 4600$ cm$^{-3}$ (56% of total number concentration) followed by the nucleation mode, with average particle number concentration of $2800 \pm 2100$ cm$^{-3}$. The contribution of the accumulation mode is the lowest in terms of particle number (19% of total number concen-

tration), but the highest in terms of particle volume concentration. The nucleation mode presented a peak concentration in the morning rush-hour, similar to eBC, $HOA_{II}$ and $NO_2$ (Fig. 5). Backman et al. (2012) identified similar diurnal cycles for nucleation-mode particles and attributed it to vehicular emissions and new particle formation (NPF) events. The nucleation mode peak in the morning is consistent with a simultaneous drop in the particle mean geometric diameter. Conversely, in the afternoon a strong increase of accumulation mode particles was observed, simultaneously with the larger presence of secondary

species such as OOA, sulfate and $O_3$. Similarly, Aitken mode particles increase during the afternoon. According to Backman




et al. (2012) the shift from nucleation to Aitken mode can be attributed to the growth of pre-existing nucleation-mode particles into the Aitken regime.

Figure 7 shows median PNSD for nucleation, Aitken and accumulation modes under low (below $25^{th}$ percentile) and high (above $75^{th}$ percentile) $PM_1$ concentrations. From this comparison, total number concentrations increase from 7000 cm$^{-3}$ (for
$PM_1 < 5.6 \mu g\ m^{-3}$) to 18400 cm$^{-3}$ (for $PM_1 > 13.7 \mu g\ m^{-3}$). Interestingly, the enhancement differs notably between aerosol size modes. The accumulation mode shows the largest increase from low $PM_1$ condition (1200 cm$^{-3}$) to high $PM_1$ condition (5200 cm$^{-3}$). On the other hand, the nucleation mode shows a smaller increase from low $PM_1$ to high $PM_1$ scenario, and therefore the contribution of the nucleation mode particles to the total number concentration is higher under low $PM_1$ (30%) than under high $PM_1$ (16%). Local traffic emissions likely remain at similar levels during both low $PM_1$ and high $PM_1$ episodes. Considering
that nucleation mode particles are mainly traffic related (Fig. 5), its relative contribution to the total number concentration tends to increase during low $PM_1$ episodes. The results are similar to Martins et al. (2010), where aerosol size distributions were measured during a transition period between a highly polluted episode and a clean one in São Paulo downtown. The authors found particles distributed in the nucleation and Aiken mode during the clean period, and larger geometric mean diameters during polluted periods, with particles partitioned between Aitken and accumulation modes. At the MASP, the meteorological
conditions that favor the occurrence of high PM concentrations are typically a low boundary layer with a low inversion layer, weak ventilation, absence of precipitation and clear sky (Sánchez-Ccoyllo and Andrade, 2002; Santos et al., 2018), favoring aerosol secondary production and particle size increase by condensation. Similarly, Rodríguez et al. (2007) attributed high ultrafine particle events to low $PM_{2.5}$ conditions in Milan, Barcelona and London. The authors also found that high $PM_{2.5}$ pollution events tend to occur when condensation processes produce significant number concentrations of accumulation mode
particles.

The strong occurrence of accumulation mode particles under polluted conditions can be explained by the fact that larger surface area of pre-existing particles favors coagulation processes. Consequently, nucleation is suppressed by coagulation loss and particles become larger. Moreover, the submicron aerosol size distribution is strongly influenced by the competition between nucleation of new particles and condensation of gas-phase components onto pre-existing particles Rodríguez et al. (2005). Un-
der polluted conditions, the aerosol surface is enough to favor the condensation of vapors onto pre-existing particles, inhibiting nucleation, and resulting in particle growth. During low $PM_1$ conditions, the available aerosol surface is low, decreasing both condensation and coagulation rates, which favors homogeneous nucleation.

### 3.4 Relationships between particle size and chemical composition of submicron particles

The contribution of aerosol size modes to the ambient concentrations of the $PM_1$ chemical species was assessed by performing
a multilinear regression model (MLR). In the MLR model, the time series of volume concentration at the nucleation, Aitken and accumulation modes were used as predictors. $PM_1$ components were used as species of interest. Results of MLR are summarized in Table 5. The model explained more than 90% of the average measured concentrations for the $PM_1$ species. For predictors used in MLR, the calculated variance inflation factor (VIF) was in the range of 1.11 to 2.16. In general, VIFs below 10 indicate no collinearity (Hair et al., 2006), ensuring the reliability of the regression results.





Contribution of aerosol size modes to mass concentrations of PM$_1$ chemical species (Figure 8) was obtained by multiplying regression coefficients (Table 5) and average volume concentrations. Their confidence intervals were calculated according to the confidence intervals of regression coefficients. PM$_1$ mass loadings were reconstructed by the sum of the partial contributions determined for each size mode (Fig. 9), i.e. nucleation (0.57 $\mu$g m$^{-3}$), Aitken (1.25 $\mu$g m$^{-3}$) and accumulation modes (6.23 $\mu$g m$^{-3}$). It could explain 75% of the mean of measured PM$_1$. It is important to emphasize that the dependent variables in the

MLR are the mass measured by ACSM and MAAP, however, the particle range measured by the ACSM goes from 70 to 900 nm, so most of the nucleation mode particle composition cannot be directly measured by the ACSM. It can result in larger uncertainties for the reconstructed mass in the nucleation mode. However, since it represents a small fraction in terms of total mass the uncertainty is likely small.

Mass loading attributed to nucleation mode is predominantly eBC and HOA factors, both related to the emission of fresh

particles from traffic. Similarly, Carbone et al. (2013) related lower diameters to the dominance of BC and organics, including typical hydrocarbons found in traffic emission (HOA). Rodríguez et al. (2007) also found a significant correlation between OA and eBC, and ultrafine particles, attributed to vehicle exhaust emissions. Besides, biofuel usage has been shown to increase the number of nucleation mode particles in the MASP (Martins et al., 2012). A substantial fraction of nucleation mode is also attributed to OOA. Photooxidation of aromatic VOCs from vehicular exhaust has been suggested to be an important driver to

NPF events under urban environments, yielding oxidized organic aerosols into nucleation mode (Gurjar et al., 2008). Moreover, Costabile et al. (2009) suggested a relation between nucleation aerosol mode in the urban atmosphere to photochemically induced particle formation, in addition to traffic emissions.

Secondary inorganic species (ammonium, nitrate and sulfate) are partitioned between Aitken and accumulation modes. Those results are similar to Rodríguez et al. (2007). The authors observed that ambient concentrations of ammonium nitrate

and ammonium sulfate correlated better with the accumulation mode, attributing it to condensation mechanisms and particle growth processes. Aitken mode looks like the most acidic size mode. Carbone et al. (2013) suggested that the presence of nitrate and ammonium in the Aitken mode is likely to result from the reaction between nitric acid (HNO$_3$) and ammonia (NH$_3$) from traffic emissions. Moreover, Backman et al. (2012) associated the growth of pre-existing nucleation mode aerosols to particle coating by sulfates and inorganic nitrates.

In this study, the BBOA presents no mass loadings in nucleation mode, a small fraction in Aitken mode, and the highest loading attributed to the accumulation mode. This result indicates that this urban site can be influenced by regional biomass burning emissions. The enhancement of accumulation mode particles under the influence of regional biomass burning emissions has been observed elsewhere, such as in Amazonia. Kumar et al. (2016) discussed the relevance of unregulated PM sources in MASP, including wood burning in pizzerias that could emit  321 kg day$^{-1}$ of PM$_{2.5}$, according to their estimates.

Submicron mass loading in accumulation mode has a great contribution of OOA (32%), probably resulting from SOA formation and aging. Accumulation mode also presents significant contributions of HOA factors (24%) and BC (21%). Similarly, Costabile et al. (2009) related the accumulation mode to particles containing aged secondary aerosol and also direct anthropogenic emissions.



## 4 Summary and conclusions

Physicochemical properties of aerosols are key parameters in terms of their impacts on human health and climate effects. In this study, a detailed characterization of submicron particles was performed at an urban site in the MASP. The results show $PM_1$ mass concentrations in close agreement with other megacities, with an average mass concentration of 11.4 $\mu$g m$^{-3}$. As expected, chemical composition was dominated by organic aerosols (56%), with significant contributions of sulfate (15%) and black carbon (13%). Using PMF analysis it was possible to identify four OA classes including oxygenated organic

aerosol (OOA), biomass burning organic aerosol (BBOA), and two hydrocarbon-like OA components (a typical HOA related to vehicular emissions, and a HOA associated to a mix of anthropogenic sources). Considering the sum of secondary inorganic aerosols and SOA as a lower limit, more than 50% of $PM_1$ mass loading was estimated as resulting from secondary production.

Nucleation, Aitken and accumulation lognormal size modes were fitted to the measured PNSD. Aitken mode dominated the total number concentration with an average concentration of 6,900 cm$^{-3}$ and submicron aerosol size distribution was strongly

influenced by the $PM_1$ levels. The accumulation mode shows a large increase from low $PM_1$ conditions to high $PM_1$ conditions, when aerosol surface is enough to favor the condensation of vapors onto pre-existing particles, inhibiting nucleation, and resulting in particle growth. Conversely, the contribution of particles from the nucleation mode to the total number concentration is higher during low $PM_1$ conditions, when the available aerosol surface is low, decreasing both condensation and coagulation rates, and favoring homogeneous nucleation. Because of the high contribution of nucleation particles under low

$PM_1$ loadings, $PM_{2.5}$ and $PM_{10}$ (parameters frequently used in the air quality index) may be insufficient to assess human PM exposure in urban areas.

The relationships between size modes and chemical constituents of $PM_1$ were assessed by performing an MLR model. Mass loading in nucleation mode was attributed to fresh particles from traffic (HOA and eBC) and photochemically induced secondary particle formation (OOA). Secondary inorganic species (ammonium, sulfate and nitrate) were partitioned between

Aitken and accumulation modes and related to condensation particle growth processes. Submicron mass loading in the accumulation mode included aged secondary organic aerosol and vehicular emissions.

The results presented here emphasize the well-established impact of traffic-related sources in the MASP and make clear the need to reduce emissions rates in the region by applying new technologies such as the EURO VI emission standard. It is also essential to expand mass transportation systems, since the metro system in São Paulo is heavily underdeveloped, resulting

in a better transportation system for 20 million people. Additionally, encouraging alternative transportation, implementing strong incentives for electrical vehicles and the restriction of passenger car circulation can improve significantly air quality in urban areas. Although the implementation of regulatory programs to control stationary and mobile sources in the MASP over the last decades has been successful to reduce primary emissions, secondary processes have been recognized as critical to air quality in the region. The findings presented provide innovative insights on the association between sources and processes

governing physicochemical properties of atmospheric aerosol and highlight the key role of SOA formation on the $PM_1$ ambient concentrations in a megacity largely impacted by traffic emissions and extensive biofuel usage.





*Data availability.* The data sets are available upon request (djunior@if.usp.br).

*Author contributions.* DMS conducted most of the data analysis and wrote the paper. PA supervised the work. LVR, PS, SC contributed to specific parts of the data analysis. DMS, LVR, PS conducted the measurements. All the authors contributed to the interpretation of the results and writing of the paper.

*Competing interests.* The authors declare that they have no conflict of interest.

*Acknowledgements.* This work was supported by the Research Foundation of the State of São Paulo (FAPESP) and by the National Council for Scientific and Technological Development (CNPq). We thank Fernando Morais, Fábio Jorge and Simara Morais for technical and logistics support in the successful operation of the sampling site. We thank the IAG Meteorological Station for providing meteorological data. DMS acknowledges scholarship from CNPq.







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





**Figure 1.** Map of Brazil indicating São Paulo state and detailed view of the sampling site (red point), located in the southeastern part of the MASP in Diadema city. Source: Esri, DigitalGlobe, GeoEye, Earthstar Geographics, CNES/Airbus DS, USDA, USGS, AeroGRID, IGN, and the GIS User Community





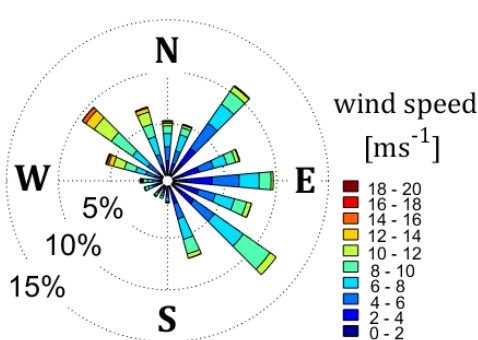

**Figure 2.** Wind rose during the campaign. Data from Institute of Astronomy, Geophysics and Atmospheric Sciences of University of São Paulo (IAG) meteorological station.





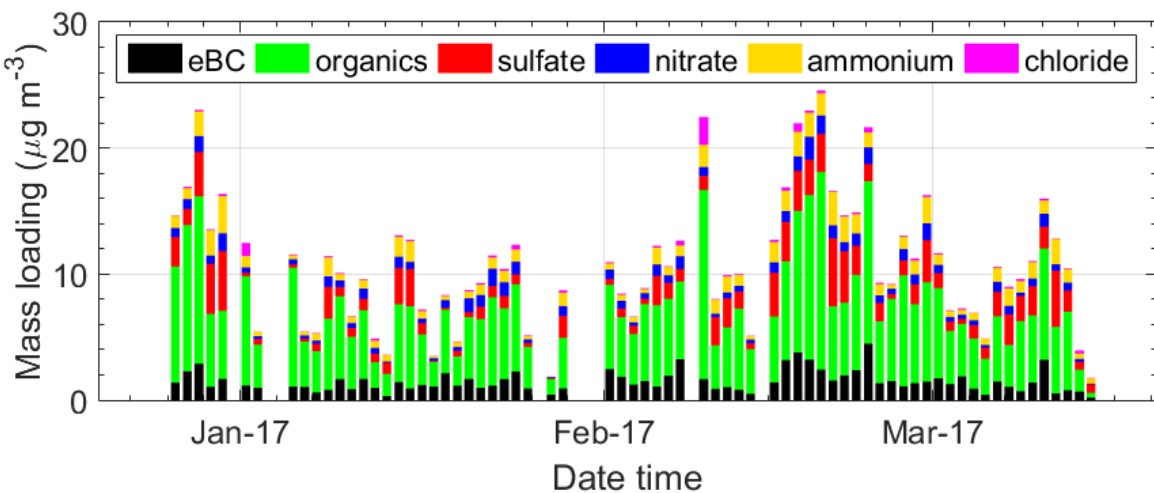

**Figure 3.** Time series of mass concentration of chemical species in PM$_1$. Each bin represents the average of measured data in 1 day intervals. The error bars represent the standard deviation of the timeseries of the sum of all components.



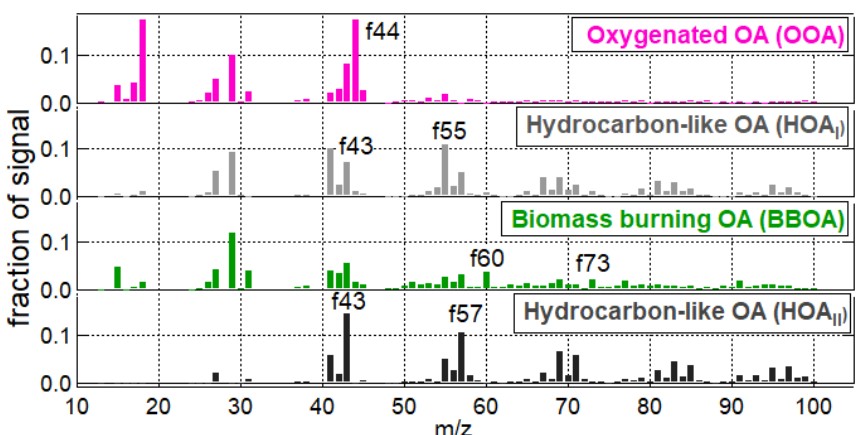

**Figure 4.** PMF mass spectra of the four PMF solutions containing oxygenated organic aerosol (OOA), biomass burning organic aerosols (BBOA) and two hydrocarbon-like organic aerosols ($HOA_I$ and $HOA_{II}$).



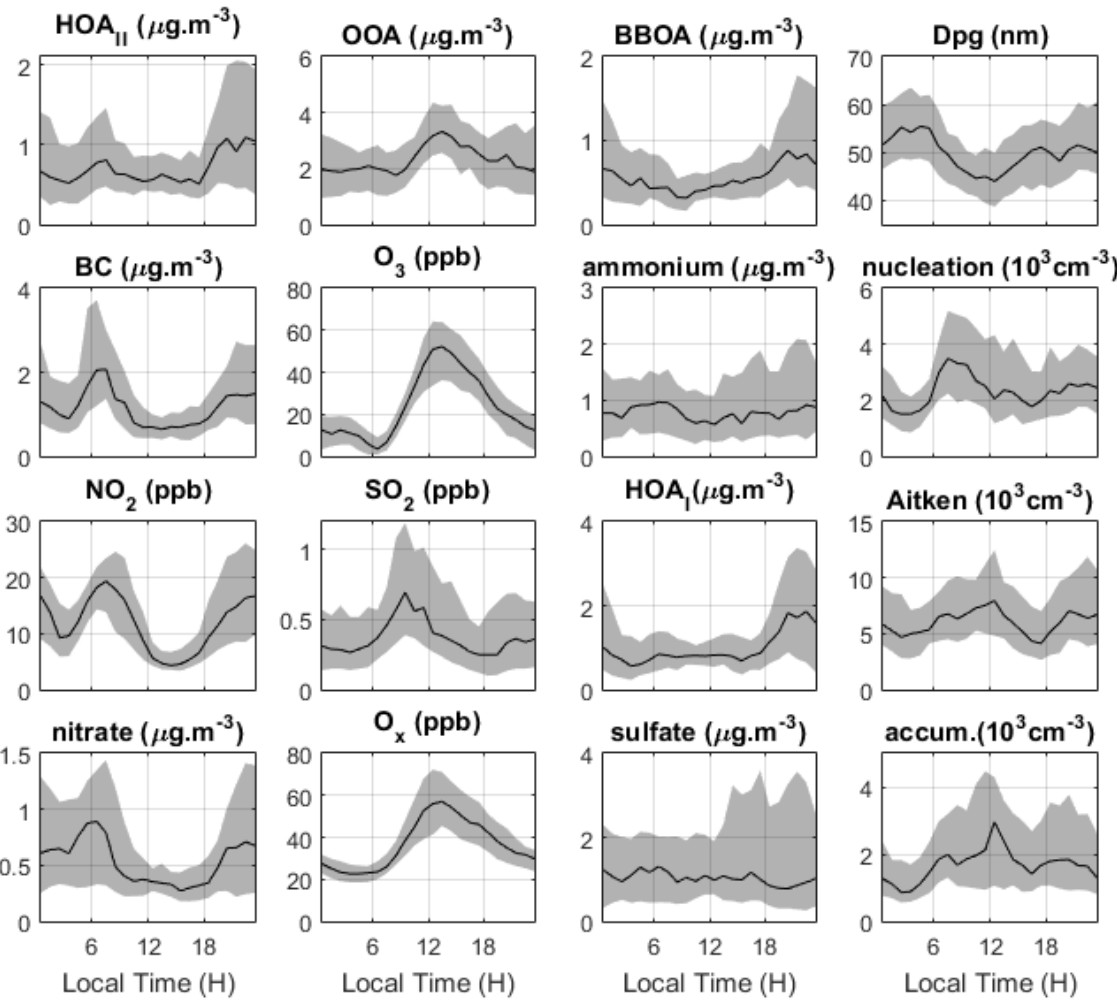

**Figure 5.** Diurnal trends for OA PMF factors (OOA, HOA$_I$, BBOA and HOA$_{II}$), black carbon, sulfate, nitrate, ammonium trace gases (NO$_2$, SO$_2$ and O$_3$), geometric mean diameter and submicrometer aerosol modes (nucleation, Aitken and accumulation). Solid line represents the median and shaded areas represent the interquartile interval.





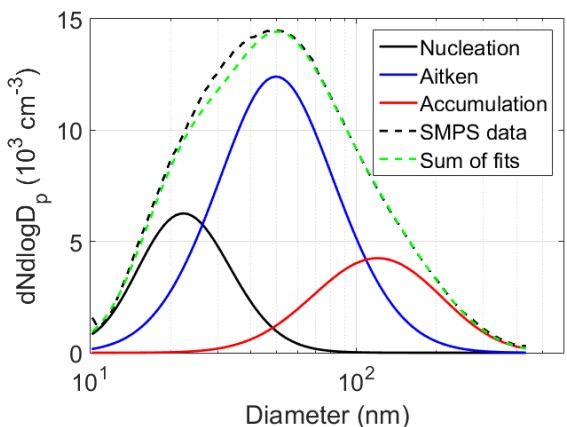

**Figure 6.** Average particle number size distribution for measured and lognormal fits for nucleation, Aitken and accumulation modes. Mean geometric diameter and geometric standard deviation of the modes are 12.6 (1.2) nm, 30.7 (1.3) nm and 83.8 (1.3) nm. The local maximum of the number distribution (mode diameter) is at 22.5 nm for nucleation, at 49.6 nm for Aitken, and at 121.9 nm for accumulation mode. The average $\pm$ standard deviation for particle number concentration is 12500 $\pm$ 7200 cm$^{-3}$



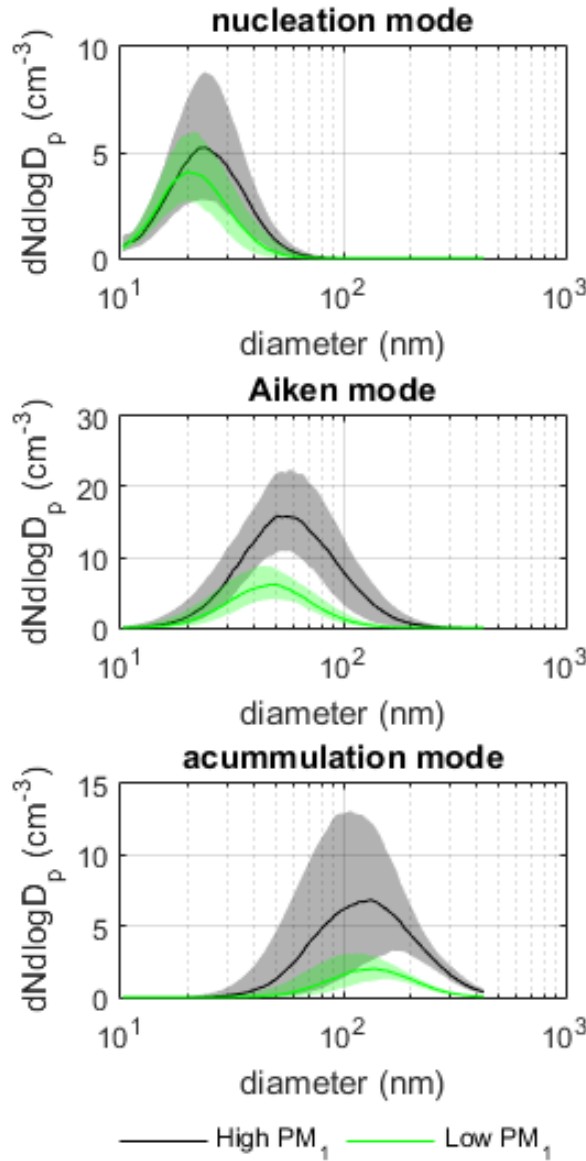

**Figure 7.** PNSD for nucleation, Aitken and accumulation modes under high (black) and low (green) PM$_1$ levels. Solid line represents median PNSD. Shaded areas represent the interquartile interval. PM$_1$ values higher than 75th percentile of time series are considered high PM$_1$ concentrations and PM$_1$ values lower than 25th percentile of time series are considered low PM$_1$ concentrations.



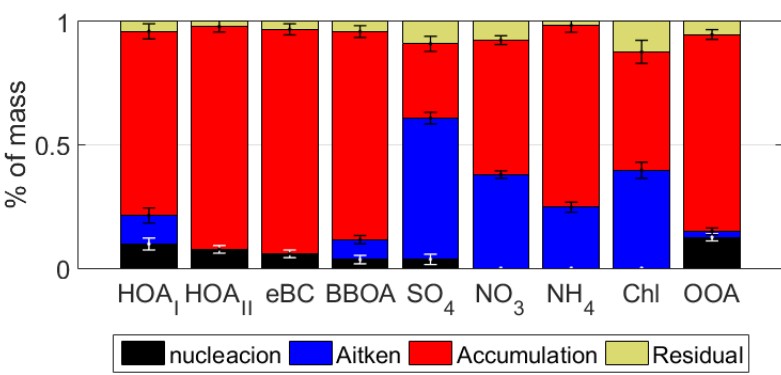

**Figure 8.** Results of MLR: contributions of aerosols size modes into mass concentrations for $PM_1$ chemical species and confidence intervals.





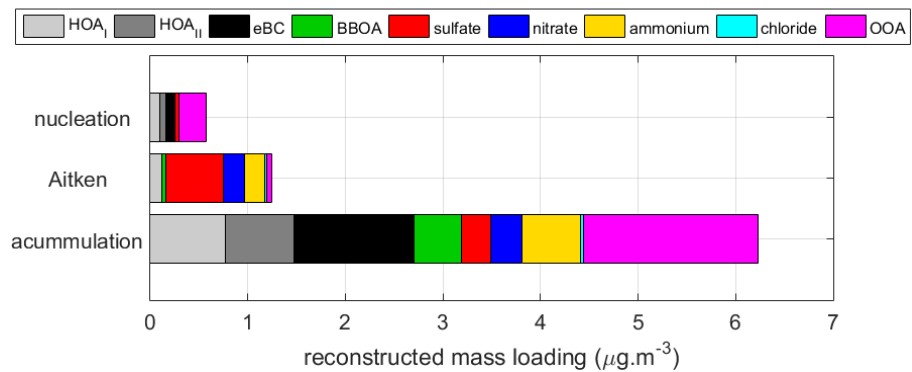

**Figure 9.** Results of MLR: reconstructed mass loading into nucleation, Aitken and accumulation modes.





**Table 1.** Summarized statistics of meteorological data and mixing ratios of trace gases from $20^{th}$ December 2016 to $15^{th}$ March 2017. Statistics includes mean concentration, standard deviation ($\sigma$), median and interquartile interval (IQ).

| variables (and units) | mean (and $\sigma$) | median (and IQ) |
|---|---|---|
| RH (%) | 79 (16) | 85 (68-92) |
| Air temperature (°C) | 23.0 (3.6) | 22.1 (20.5-25.4) |
| mid-day $O_3$ (ppb) | 43.5 (19.8) | 40.6 (29.0-55.0) |
| $NO_2$ (ppb) | 13.2 (8.9) | 11.2 (5.9-18.8) |
| $SO_2$ (ppb) | 0.61 (0.85) | 0.36 (0.19-0.67) |





**Table 2.** Summarized statistics of near real-time aerosol measurements (PM$_1$ and its individual constituents) from $20^{th}$ December 2016 to $15^{th}$ March 2017. Statistics includes mean concentration, standard deviation ($\sigma$), median and interquartile interval (IQ).

| variables (and units) | mean (and $\sigma$) | median (and IQ) |
|---|---|---|
| PM$_1$ ($\mu$g m$^{-3}$) | 11.4 (7.8) | 9.7 (5.9 - 14) |
| eBC ($\mu$g m$^{-3}$) | 1.5 (1.4) | 1.1 (0.6 - 1.9) |
| organics ($\mu$g m$^{-3}$) | 6.3 (5.0) | 5.1 (3.2 - 7.7) |
| sulfate ($\mu$g m$^{-3}$) | 1.6 (1.8) | 1.02 (0.42 - 2.30) |
| nitrate ($\mu$g m$^{-3}$) | 0.70 (0.67) | 0.46 (0.24 - 0.98) |
| ammonium ($\mu$g m$^{-3}$) | 1.03 (0.92) | 0.77 (0.08 - 1.50) |
| chloride ($\mu$g m$^{-3}$) | 0.19 (0.34) | 0.05 (0.01 - 0.13) |





**Table 3.** Pearson correlation coefficients (R) between mass spectra of OA PMF factors and AMS database (http://cires1.colorado.edu/jimenez-group/AMSsd/).

| Pearson correlation (R) with AMS database | OOA | HOA I | BBOA | HOA II |
|---|---|---|---|---|
| HOA (Ng et al., 2011b) | 0.26 | 0.90 | 0.53 | 0.89 |
| SV-OOA (Ng et al., 2011b) | 0.72 | 0.68 | 0.70 | 0.58 |
| LV-OOA (Ng et al., 2011b) | 0.99 | 0.30 | 0.41 | 0.10 |
| OOA (Ng et al., 2011b) | 0.98 | 0.30 | 0.41 | 0.10 |
| BBOA (Ng et al., 2011b) | 0.64 | 0.79 | 0.90 | 0.52 |





**Table 4.** Pearson correlation coefficients (R) between time series of OA factors and time series of aerosol components and trace gases. Aerosol components include sulfate, nitrate, ammonium, chloride and equivalent black carbon time series. Trace gases include sulphur dioxide, nitrogen dioxide, ozone and oxidant concentration (Ox=$O_3$+$NO_2$) time series.

| Pearson correlation (R) with external tracers | OOA | HOA I | HOA II | BBOA |
|---|---|---|---|---|
| $O_X$ | 0.48 | -0.06 | -0.07 | -0.03 |
| sulfate | 0.46 | 0.04 | 0.04 | 0.15 |
| $SO_2$ | 0.35 | 0.08 | 0.07 | 0.08 |
| ammonium | 0.53 | 0.16 | 0.19 | 0.30 |
| nitrate | 0.63 | 0.43 | 0.58 | 0.48 |
| eBC | 0.40 | 0.45 | 0.69 | 0.47 |
| $NO_2$ | 0.23 | 0.55 | 0.60 | 0.31 |
| $O_3$ | 0.36 | -0.23 | -0.28 | -0.14 |
| chloride | 0.19 | 0.23 | 0.32 | 0.58 |



**Table 5.** Fit parameters of MLR model results between $PM_1$ components (species of interest) and volume of aerosol size modes (predictors) and adjusted $R^2$. [a] Regression coefficients with pValues> 0.1, not statistically significant in a 90% confidence interval. [b] Regression coefficients with $0.05 < p < 0.1$.

| $PM_1$ species | $\beta$ nucleation | $\beta$ Aitken | $\beta$ accumulation | Adj. $R^2$ |
|---|---|---|---|---|
| nitrate | 0.25 (0.20)[b] | 0.139 (0.005) | 0.042 (0.001) | 0.74 |
| OOA | 7.44 (0.87) | 0.034 (0.019)[b] | 0.239 (0.006) | 0.64 |
| eBC | 2.09 (0.56) | -[a] | 0.164 (0.004) | 0.62 |
| $HOA_{II}$ | 1.56 (0.32) | -[a] | 0.093 (0.002) | 0.59 |
| sulfate | 0.96 (0.57)[b] | 0.368 (0.015) | 0.041 (0.004) | 0.58 |
| BBOA | 0.52 (0.26) | 0.029 (0.006) | 0.064 (0.002) | 0.56 |
| ammonium | -[a] | 0.128 (0.011) | 0.080 (0.003) | 0.54 |
| $HOA_I$ | 2.68 (0.66) | 0.08 (0.02) | 0.103 (0.004) | 0.45 |
| chloride | -[a] | 0.015 (0.001) | 0.004 (0.001) | 0.34 |