# Peer review of "Physical and chemical properties of urban aerosols in São Paulo, Brazil: Links between composition and size distribution of submicron particles."

_Atmospheric Chemistry and Physics, 2020_

## Author Comment (AC1)

**Response to Referee #1 and  Referee #2 comments**

We thank the two referees that made very good and constructive comments in our manuscript. It certainly helped a lot in getting a much better final manuscript. In particular, the comments helped to improve the multiple linear regression model, helping to avoid potential issues with the nucleation mode and the chemical composition measured by the ACSM. Also general comments have improved small errors such as units, and plots. We implemented all suggestions done by the two reviewers.

In order to organize all individual comments and answers plus actions, we highlighted the comments and answers by the following way: Blue text shows the referee #1 comment [R1] and the referee #2 comment [R2], black text shows the authors' response [A]. Modified and new figures are presented at the end of this document.

**Referee #1 Comments and authors' response and actions**

**[R1 General Comment]:** *The study investigated the physical and chemical properties of submicron aerosols in Sao Paulo, Brazil mainly based on the real-time measurements by aerosol chemical speciation monitor and scanning mobility particle sizer. The composition and sources of organic aerosol (OA) were analyzed with positive matrix factorization, and the results showed four OA factors relating to different sources and processes. A major highlight of this study is linking aerosol composition to the size distributions, showing different contributions of aerosol species to nucleation, Aitken and accumulation mode. The results are important for a better understanding of the impacts of different sources on air quality in metropolitan area of Sal Paulo. This manuscript is overall well written, and I have a few comments.*

**[A1]** We thank the positive evaluation of the manuscript by the reviewer.

**[R1.1]** *Considering the authors have the measurements of both submicron aerosol species and size-resolved particle number concentrations, I suggest the authors comparing the mass measured by ACSM and MAAP with the volume or mass estimated from SMPS to validate the quality of the data.*

**[A1.1]** As suggested by the referee, we include the comparison between the mass measured by ACSM  and MAAP with volume estimated from SMPS, as a new figure in the supplementary material (Figure S6). Result of intercomparison shows a very good correlation between PM1 mass concentration and Volume estimated from SMPS ($R^2$=0.89, p<0.001).

**[R1.2]** *I suggest combining Figure 1 and Figure 2.*

**[A1.2]** Thanks for the suggestion, we combine Figure 1 and Figure 2 (now Figure 1). The map together with the wind direction now is on the same figure that facilitates the visualization.

**[R1.3]** *Figure 3 did not show any error bars.*

**[A1.3]** Thanks for observing this omission. In the revised version, we include error bars in Figure 3 (now Figure 2).

**[R1.4]** *The labels of fm/z in Figure 4 are not appropriate, for example, f44 is the fraction of m/z 44 in OA, it is better use 44.*

**[A1.4]** We agree that $f_{m/z}$ is not appropriate in this context. We use "44" instead of "$f_{44}$" in Figure 4 (now Figure 3).

**[R1.5]** *Please check the unit of SO2 in the text. It was "ppm" sometimes.*

**[A1.5]** Sorry for our mistake. In the revised version we fix it in the text, as below:

> *"The average $SO_2$ concentration is 0.61 ppb, approximately half of the observed values in São Paulo downtown during springtime in 2013 (Monteiro dos Santos et al., 2016). Few episodes (15% of the dataset) presented $SO_2$ levels above 1 ppb."*

**[R1.6]** *Figure 7, please check the scale of y-axis, missed "103"?*

**[A1.6]** Thanks for noting that, we fix the scale of the y-axis ($10^3$) in Figure 7 (now Figure 6).

**2.2 Referee #2 Comments and authors' response**

**[R2 General Comment]:** *This manuscript reported the composition of submicron aerosol measured by an aerosol chemical speciation monitor (ACSM) in Sao Paulo, Brazil. PMF analysis is performed for the sources apportionment of organic aerosol. One novel aspect of this manuscript is to apply multivariate linear regression (MLR) analysis to estimate the size-dependent PM composition. However, several concerns regarding this analysis are listed below and should be carefully addressed. Given that the majority of the manuscript focuses on reporting the measurements of PM composition and size distributions, I recommend this manuscript to be classifies as measurement report. Overall, I recommend publication after major revisions.*

**[A2 - Author Response]:** We thank referee #2 for critical feedback and constructive suggestions. We agree with concerns regarding MLR analysis and hence we recalculate the MLR analysis taking into account the reviewer's comments. Regarding the reviewer recommendation to classify the manuscript as a measurement report, we emphasize that this work analyzes the aerosol chemical composition that is normally integrated into a

large size range, into modes of aerosol size distribution. It provides separation of chemical components into accumulation and Aitken modes. We believe that the attribution of the chemical composition with size-dependent PM provides innovative information on the properties of both primary and secondary organic aerosols, as well as inorganic aerosols in a complex urban environment.

**[R2.1]** *One major concern regarding the MLR analysis is that the mass balance seems violated. In Table 5, the sum of beta coefficients over one column should be no larger than 1, in order for the coefficients to be physically meaningful. In addition, the MLR model results of each PM constituent are additive, which leads to total PM1 concentration = 15.5 \* nucleation model volume + 0.793 \* Aiken model volume + 0.826 \* Accumulation model volume. This suggests that more particles in the nucleation model are required by MLR than actual measurement. Another way to check the mass closure is to convert the reconstructed mass loading in each mode to volume using particle density and then compare the converted volume to the measured volume in that mode.*

**[A2.1]** Yes, this effect is caused by the fact that the nucleation mode (local maximum at 22.5 nm) is out of the particle size range measured by the ACSM (75–650 nm), as pointed out by the referee. Taking this into account, we modified the model, now excluding the nucleation mode in the MLR model. Since the nucleation mode represents only a small fraction of the aerosol volume (<0.5%) and hence of the mass concentration, it is reasonable to explain the PM mass concentration based on the Aitken and accumulation modes. Results were similar to the previous calculations for Aitken and accumulation (new Figure 7 and Figure 8), with slightly higher residuals and no violation in mass balance. We also included in the discussions at the revised version the reasons for not using nucleation mode in MLR analysis.

**[R2.2]** *Another concern is that particle sizes measured by ACSM (70-900 nm aerodynamic diameter) and by SMPS (10-450 nm electrical mobility diameter) do not fully overlap. As large particles account for a dominant fraction of PM mass, the narrower range of SMPS measurement may cause substantial uncertainty in the MLR analysis.*

**[A2.2]** Yes, it is important that both instruments measured a similar size range to reduce uncertainties in the MLR analysis. We corrected the listed ACSM size range. According to Ng et al. 2011 the aerodynamic lens used in the Q-ACSM, has a high transmission efficiency in the aerodynamic diameter range 70–500 nm (Canagaratna et al. 2007 and references therein). Considering the mobility size range measured by the SMPS (10-450nm), the particles measured by the SMPS in vacuum aerodynamic diameter is 12 – 540 nm, which comprise the Aitken and Accumulation modes, and both instruments have similar upper particle measurement capabilities. The accumulation mode aerosol particles in this study were centered on 122 nm (mobility diameter) and 146 nm (vacuum aerodynamic diameter), suggesting that both instruments fully measured this mode.

**[R2.3]** *More information about the procedure to fit the particle size distribution should be provided. How to decide the number of lognormal modes in fitting the measured particle*

*size distribution? Is the center of each mode allowed to vary between different particle size distributions? If so, what is the variation of the center diameter?*

**[A2.3]** We agree with the referee that the manuscript needs more information about the fitting procedure. We include the following sentence in line 196:

*"An algorithm in Matlab was developed to fit one to three lognormal modes to the measured particle number size distributions, based on Hussein et al. (2005), using the standard method of least-squares. The center of each mode is allowed to vary from one size distribution to another, constrained to the following diameter ranges: nucleation mode mean geometric diameter (Dpg) must be smaller than 30 nm; Aitken mode Dpg must be ≥ 30 nm and < 90nm; accumulation mode Dpg must be ≥90 nm. There is also a restriction in the geometric standard deviation, which must be within 1.2 and 2.1 for all modes. The algorithm automatically decides between 2 or 3 mode fitting based on the percentual concentration of 10-40 nm particles, and based on goodness of fit proxies like the root mean square error and the ratio between fit and data total particle number concentration. The algorithm is able to automatically reduce the number of fitted modes (from 3 to 2 modes and from 2 to 1 mode) if a complete superposition is detected, based on the ratio between the modes Dpg. The algorithm uses the previous fitting parameters as a first guess to the current fitting, so that continuity is favored."*

**[R2.4]** *The discussions in section 3.3 is scientifically correct, but cannot be inferred based on the observations in this study. In this section, the particle size distributions are compared between low and high PM concentration. As the PM concentration is driven by large particles, it is not surprising that the number concentration in accumulation mode shows the largest difference in this comparison, followed by aiken mode. Looking at Lines 290-292 in conclusion section, it is correct that "the accumulation mode shows a larger increase from low PM1 conditions to high PM1 conditions". In this same line, it is also correct that "when aerosol surface is enough to favor the condensation of vapors onto pre-existing particles, inhibiting nucleation, and resulting in particle growth". However, the linkage between these two correct statements are missing in this study. In other words, what is observed in this study is due to how the data are segregated, and it cannot be used to infer any particle growth mechanism.*

**[A2.4]** We agree that the linkage between these two correct statements are missing in this study. In the revised version, we adapted the sentence in lines 290-292 as follow:

*"The accumulation mode shows a large increase from low PM1 conditions to high PM1 conditions. The shift in the particle size distribution to larger sizes provides more aerosol surface, which can favor the condensation of vapors onto pre-existing particles and likely inhibit nucleation. It corroborates with lower contribution of particles at nucleation mode observed under high PM1 conditions."*

**[R2.5]** *Results from the MLR analysis should be elaborated and further explored. For example, why is eBC dominantly in the accumulation mode, but almost zero in the Aiken mode? Does this observation suggest the eBC at the measurement site is heavily coated from regional transport? If so, does it contradict with vehicle emission as the major*

*source in this study? Why is a large fraction of inorganic ions in the Aiken model, but not the case for organics?*

**[A2.5]** We agree with the referee that the MLR results should be explained in more detail. In the revised version, we adapted the sentence in lines 263-269, as follow:

*"Secondary inorganic species (ammonium, nitrate and sulfate) are partitioned between Aitken and accumulation modes. Similarly, Rodríguez et al. (2007) observed strong correlation between ammonium nitrate and ammonium sulfate with the accumulation mode, attributing it to condensation mechanisms and particle growth processes. A large fraction of inorganic species are in the Aitken mode, and it looks like the most acidic size mode. Carbone et al. (2013) suggested that the presence of nitrate and ammonium in the Aitken mode is likely to result from the reaction between nitric acid ($HNO_3$) and ammonia ($NH_3$) from traffic emissions. Moreover, Backman et al. (2012) associated the growth of pre-existing nucleation mode aerosols to particle coating by sulfates and inorganic nitrates."*

We also include the following sentence in lines 278:

*"Interestingly, most of eBC concentration is in the accumulation mode and only a small fraction of mass is in Aitken mode. The same behaviour is observed for traffic related HOA factors. It can indicate that a substantial fraction of the BC is heavily coated by organics from regional transport, in addition to local vehicular fresh emissions, however it needs further investigations. A suggestion for future studies is the analysis of BC mixing state and BC coating thickness using a single-particle soot photometer (SP2)."*

**[R2.6]** *Stepwise selection should be applied in the MLR to explore the explanatory power of each term. In table 5, please list the p value of each term, which will help to examine the importance of each predictor.*

**[A2.6]** We thank the referee for the suggestion. Since we recalculate the MLR model using Aitken and accumulation mode as predictors, we now also include the p value of each individual calculated value in table 5.

**[R2.7]** *Line 134. Is the SO2 0.61 ppm or ppb?*

**[A2.7]** Thanks for the correction, we fix it in the text, as below:

"The average $SO_2$ concentration is 0.61 *ppb*, approximately half of the observed values in São Paulo downtown during springtime in 2013 (Monteiro dos Santos et al., 2016). Few episodes (15% of the dataset) presented $SO_2$ levels above 1 *ppb*."

**Modified figures.**

[Figure]

**Figure S6** - Comparison of the $PM_1$ mass concentration (ACSM plus MAAP) with the aerosol volume concentration derived from the SMPS.

[Figure]

**Figure 1.** Map of Brazil indicating São Paulo state and detailed view of the sampling site (red point), located in the southeastern part of the MASP in Diadema city. Source: Esri, DigitalGlobe, GeoEye, Earthstar Geographics, CNES/Airbus DS, USDA, USGS, AeroGRID, IGN, and the GIS User Community. Wind rose during the campaign was calculated from data from Institute of Astronomy, Geophysics and Atmospheric Sciences of University of São Paulo (IAG) meteorological station.

[Figure]

**Figure 3 (now Figure 2).** Time series of mass concentration of $PM_1$ chemical species. Each bin represents 1 day-average average of measured data. The error bars represent the standard deviation of the sum of all components.

[Figure]

**Figure 4 (now Figure 3).** PMF mass spectra of the four PMF solutions containing oxygenated organic aerosol (OOA), biomass burning organic aerosols(B BOA) and two hydrocarbon-like organic aerosols (HOA$_I$ and HOA$_{II}$).

[Figure]

**Figure 7 (now Figure 6).** PNSD for nucleation, Aitken and accumulation modes under high (black) and low (green) PM$_1$ levels. Solid line represents median PNSD. Shaded areas represent the interquartile interval. PM$_1$ values higher than 75[th] percentile of time series are considered high PM$_1$ concentrations and PM$_1$ values lower than 25[th] percentile of time series are considered low PM$_1$ concentrations.

[Figure]

**Figure 8 (now Figure 7).** Contributions of aerosols size modes into mass concentrations for PM1 chemical species and confidence intervals. Results from the MLR model.

[Figure]

**Figure 9 (now Figure 8).** Reconstructed mass loading into Aitken and accumulation modes, as attributed by the MLR model.